# Health Care Providers’ Perspectives on Promoting Physical Activity and Exercise in Health Care

**DOI:** 10.3390/ijerph19159466

**Published:** 2022-08-02

**Authors:** Liam P. Pellerine, Myles W. O’Brien, Chris A. Shields, Sandra J. Crowell, Robert Strang, Jonathon R. Fowles

**Affiliations:** 1Division of Kinesiology, Dalhousie University, Halifax, NS B3H 4R2, Canada; liam.pellerine@dal.ca (L.P.P.); myles.obrien@dal.ca (M.W.O.); 2Centre of Lifestyle Studies, School of Kinesiology, Acadia University, Wolfville, NS B4P 2K5, Canada; chris.shields@acadiau.ca; 3Nova Scotia Health Authority, Halifax, NS B3H 1V8, Canada; bailliecrowell@eastlink.ca; 4Department of Health and Wellness, Government of Nova Scotia, Halifax, NS B3J 1V9, Canada; robert.strang@novascotia.ca

**Keywords:** self-reflection survey, prescribing exercise, community-based programs, collaborative health care, continuing education

## Abstract

Health care providers (HCPs) are entrusted with providing credible health-related information to their patients/clients. Patients/clients who receive physical activity and exercise (PAE) advice from an HCP typically increase their PAE level. However, most HCPs infrequently discuss PAE or prescribe PAE, due to the many challenges (e.g., time, low confidence) they face during regular patient care. The purpose of this study was to ascertain HCPs’ perspectives of what could be done to promote PAE in health care. HCPs (*n* = 341) across Nova Scotia completed an online self-reflection survey regarding their current PAE practices and ideas to promote PAE. The sample consisted of 114 physicians, 114 exercise professionals, 65 dietitians, and 48 nurses. Quantitative textual analysis (frequency of theme ÷ number of respondents) was performed to identify common themes to promote PAE in health care. In the pooled sample, the primary theme cited was to increase the availability of community programs (24.1% of respondents), followed by more educational opportunities for providers (22.5%), greater promotion of PAE from HCPs (17.1%), reducing financial barriers experienced by patients/clients (16.3%), and increasing availability of qualified exercise professionals (15.0%). Altogether, increased PAE education and greater availability of affordable community PAE programs incorporating qualified exercise professionals, would reduce barriers preventing routine PAE promotion and support the promotion of PAE in Nova Scotia.

## 1. Introduction

For decades, one of Canada’s largest public health issues has been dealing with the continual rise in the prevalence of chronic disease and multimorbidity [1,2]. Research has shown that regular physical activity and exercise (PAE) are effective as medicinal drugs for treating a variety of chronic diseases, simultaneously [3,4]. Nova Scotia is a province located on the east coast of Canada with a population of just under 1 million people. The importance of promoting routine PAE is heightened in Nova Scotia, as the province has a higher incidence of obesity, cancer, type II diabetes, and heart disease, compared to the rest of Canada [5]. Having health care providers educate patients and clients about PAE, counsel patients and clients on PAE, and prescribe PAE to patients and clients effectively increases PAE levels [6]. A systematic review of the perspectives and practices of various health care providers (e.g., physicians, nurses, dietitians, exercise professionals) to promote PAE determined that, although 86% of health care providers believe PAE is important, only 42% routinely assess for PAE levels [7] and few frontline health care providers (i.e., physicians, dietitians, nurses) recommend PAE as part of their clinical care, citing numerous barriers (e.g., lack of time, inadequate training, disinterest from patients and clients) and low self-confidence in providing PAE prescriptions [8].

Health care providers appear to discuss PAE, provide PAE recommendations, and write PAE prescriptions to patients and clients at different rates [7]. For instance, a survey of 13,166 Canadian physicians determined 85% reported talking about PAE, but only 16% of physicians prescribed PAE to patients [9]. In an open-ended survey of physicians, respondents mentioned time restraints, inexperience in providing quality PAE advice, negative mindsets of patients, and lack of cost effective PAE programs as barriers to providing effective PAE counseling [10]. Similarly, 52% of Nova Scotian dietitians regularly provide PAE information to clients but, like physicians, only 16% routinely prescribe PAE [11]. An open-ended survey of dietitians and specialized dietary nurses identified insufficient training, not enough exercise professionals to refer to, and time restrictions as the main deterrents to PAE promotion [12]. In contrast, physiotherapists in Nova Scotia provide PAE information in 97% of appointments and regularly provide PAE advice to most clients, but still cite barriers, such lack of patient interest, lack of time and lack of resources for clients with chronic disease [13].

The lack of comfort to include PAE information by providers and sporadic inclusion of PAE in health care encounters is well established [7,9,10,12,14]. There is a clear need to identify strategies that could be implemented to address this lack of physical activity promotion in health care. In a region struggling with access to physicians, diversifying regular health care to include additional health care providers, such as exercise professionals, nurses, and dietitians, through physician referrals could be a realistic solution [15]. As of April 2022, at least 8.8% of Nova Scotians do not have access to a primary care physician [16]. Gathering the perceptions of health care providers regarding how PAE could be a regular part of their clinical care may provide insight as to how researchers, policy makers, and other decision makers could intervene. While physicians are often targeted for physical activity promotion [9], a multi-disciplinary approach to addressing physical inactivity that includes additional health care providers that have frequent patient contact (e.g., dietitians, nurses, physiotherapists, etc.) is needed [17].

The purpose of this study was to ascertain the perspectives of health care providers in Nova Scotia regarding what could be done to promote and sustain PAE in health care settings. This descriptive study analyzed separate thematic responses for physicians, nurses, dietitians, and exercise professionals, to solicit perspectives across a variety of health care professions.

## 2. Materials and Methods

### 2.1. Participants

Primary care providers and allied health professionals across Nova Scotia (*n* = 596) completed an online province-wide survey on PAE practices (FluidSurveys, Ottawa, ON, Canada) to identify the state of physical activity counselling and exercise prescription in health care. Primary care providers were defined as physicians, nurse practitioners, nurse specialists, or physician assistants who provided access to a variety of health care services [18]. Allied health professionals, such as dietitians, occupational therapists, physical therapists, and exercise professionals, have unique skill sets to provide additional access to health care services [18]. The survey was created through an initiative of the Nova Scotia Health Authority Office of Research and Innovation, in collaboration with the provincial health professional associations and prominent disease awareness and prevention agencies. The Nova Scotia Health Authority, provincial professional associations, and provincial diabetes and cancer care programs endorsed the survey and distributed it via their employee emails and their online newsletters. Informed consent was obtained from all individual participants included in the study. Only participants that responded to the open-ended question, “What could be done to promote and sustain physical activity and exercise in health care?” were included in the present study. This included physicians (*n* = 114), nurses (*n* = 48), dietitians (*n* = 68), and exercise professionals (*n* = 114). Most physicians were family physicians (*n* = 72), with some specialist physicians (*n* = 42). Nurses were either registered nurses (*n* = 39) or nurse practitioners (*n* = 8). Most exercise professionals were physiotherapists (*n* = 98), with some chiropractors (*n* = 13) and kinesiologists (*n* = 3), although all worked within the Nova Scotia Health Authority to provide exercise prescription within their scope of practice, so were grouped together as ‘exercise professionals’. The physical activity counselling and exercise prescription practices of Nova Scotia physicians [19], physiotherapists [13], and dietitians [11] have been previously presented. The analyses, purpose, and information presented in the current study are unique and used to answer an independent research question.

### 2.2. Provider Physical Activity Perception Questionnaire

Self-reflection questionnaires were adapted from previously used measures of activity prescription [8,9,20,21] as described in more detail by [22]. The survey was beta-tested, edited, and approved by the Exercise is Medicine Nova Scotia Steering Committee. In brief, the self-reflection survey included: demographics, practice history, confidence, and barrier impact. Practice history was collected using visual analog scales from 0–100%. Practice history was presented to characterize the sample but was not discussed in detail as the provider specific perceptions and practices have been presented in detail elsewhere [11,13,19]. An example presentation of questions were presented in [13]. The survey ascertained open-ended responses to the question, “*What could be done to promote and sustain physical activity and exercise in health care*?”. Importantly, no examples or prompts were provided to guide responses. Open-ended responses were coded into categories from frequently cited statements. Two reviewers independently reviewed the open-ended responses, generated themes for each provider, and met to discuss inconsistencies. Each response was characterized into meaning units. Example of actual response, “More community-based free or affordable classes. Make it a ‘code’ for billing or shadow billing as it is more important than many other things we do”. This phrase was coded as having three meaning units: (1) availability of a community program, (2) reduce patient financial barriers, and (3) improved billing structure. Each provider perspective could have multiple meaning units but duplicates of each theme within one provider was coded as one (e.g., if ‘more tools for providers’ was mentioned twice, this theme was counted once). Descriptive statistics (mean, standard deviation, and proportion) were calculated on participant characteristics and practice variables.

## 3. Results

The participant characteristics across the health care professions are presented in Table 1. As expected, exercise professionals frequently included physical activity content (85 ± 23%) and prescribed exercise (83 ± 24%) to their patients (Table 1). Physicians (13 ± 21%), dietitians (20 ± 29%), and nurses (19 ± 30%) infrequently prescribed exercise. Referrals to community exercise programs or to exercise professionals were also sporadic, as only 10 ± 18% of physicians, 16 ± 22% of dietitians, 17 ± 27% of nurses and 25 ± 26% and exercise professionals reported doing so (Table 1). Most dietitians (62%) reported at least some exercise-related educational training (e.g., university course, training workshop, webinar, etc.), but fewer physicians (30%) and nurses (37%) reported any previous exercise-related educational training.

The pooled sample generated 579 meaning units (1.7 responses per provider) with the most common themes being: availability of community programs (24%), more education for health care providers (23%), greater promotion by health care providers (17%), reduced financial barriers (16%), and availability of qualified exercise professionals (15%) (Figure 1). Other, less common, themes in the pooled sample were greater public promotion (11% of providers), tools/resources for health care providers (11%), intervention in childhood (9%), better billing structure (9%), and tools/resources for patients/clients (7%).

The proportion of meaning units did not vary by profession: physicians (189 units/114 respondents; average: 1.7 units/physician), exercise professionals (201/114; 1.8 units/exercise professional), dietitians (103/65; 1.6 units/dietitian), and nurses (87/48; 1.8 units/nurse). A summary of the top 3 themes across each of the professions are presented in Figure 2, with the detailed results presented in Table 2. The availability of community exercise programs was among the top themes across all four professions (21% of physicians, 33% of exercise professionals, 28% of dietitians, and 29% of nurses), and greater educational opportunities were a top theme among physicians (20%), dietitians (40%), and nurses (25%). Physicians were the only profession to cite changes in billing structure as a top theme (21% of physician respondents) and nurses were the only profession to strongly cite (31% of nurse respondents) reductions in patient financial barriers (Figure 2).

## 4. Discussion

The purpose of this study was to ascertain the opinions of health care providers in Nova Scotia regarding what could be done to promote and sustain PAE in health care. The two most cited themes across the pooled sample of survey respondents were: (1) greater availability of community exercise programs to refer their patients and clients to, and (2) more educational opportunities to help train health care providers on how to promote activity to their patients and clients. While these themes were among the most cited within each professional group, our study also identified some professional-specific themes, such as changes in billing structure for physicians and reducing patient financial barriers from nurses. The perceptions regarding how to best integrate PAE in health care across different health care professions who have a lot of patient contact may help direct much-needed efforts in helping more Nova Scotians lead physically active lifestyles.

Health care providers prescribing aerobic exercise increased PAE levels by ~10% at 12-month follow-up [23] and reduced risk of all-cause mortality by 30% over 12-months [24]. The term ‘exercise prescription’ may be interpreted differently across health care providers. For example, physicians may interpret ‘prescription’ as a regulated service for providing authorization for patients to receive regulated pharmaceutical products. For other providers, who do not generally prescribe regulated pharmaceutical products, prescription may have a health care provider-specific meaning. For our study, the term ‘exercise prescription’ was providing a written exercise recommendation to patients and clients (e.g., achieve at least 150 min of moderate-to-vigorous physical activity every week). To address the underutilization of PAE promotion and prescription, most health care providers agree there is a gap between the development and implementation of PAE resources. Tools/resources for either patients/clients or health care providers were mentioned by ~1 in 10 health care providers, and the call for more educational training was mentioned by 20% of physicians, 23% of exercise professionals, 40% of dietitians, and 25% of nurses. Previous work from our group demonstrated physicians with PAE training and high confidence in PAE counseling were more likely to prescribe PAE [19]. For existing health care providers, creating more opportunities to engage in continuing education opportunities to bolster their confidence in PAE counselling and increase awareness of existing tools could be an effective solution. Most (85%) of Canadian medical students do not feel they are adequately trained to prescribe PAE with 89% requesting additional training on effective activity promotion strategies [25]. However, expanding the curriculum of the training programs outside exercise professions for various frontline health care providers (i.e., physicians, dietitians, nurses) could improve the rates of PAE promotion and prescription in Canada going forward.

There are internal (e.g., intrinsic motivation) and external factors (e.g., amount of leisure time) that impact the lifestyles of patients/clients. As such, providing individually tailored exercise counseling and prescriptions may be a more impactful intervention than brief recommendations to facilitate patients leading more physically active lifestyles, via self-confidence in their ability to exercise and address external barriers (e.g., time, money, etc.). Many respondents identified the desire to have more community exercise programs or greater availability of exercise professionals that have the ‘expertise’ or ‘appropriate training’ to help them. From this, we characterized an availability theme using the term *qualified* exercise professional. In all respects, *qualified* exercise professionals are those that have the appropriate education (including post-secondary education in exercise sciences and an advanced certification in the area), scope of practice, and liability insurance to work with individuals with medical conditions [26]. Therefore, this umbrella term includes exercise professionals already identified that work in health care (such as physiotherapists, chiropractors, or kinesiologists), but may also include exercise professionals working in the community, such as affiliated kinesiologists, clinical exercise physiologists, and personal trainers with supplemental training and/or appropriate certifications (i.e., certifications developed by experts in the field which includes theory and/or practical evaluation of content). While discussion of the regulations for exercise professionals is beyond the scope of this paper, we interpreted the theme of ‘availability of a *qualified* exercise professional’ as the need for health care providers to have confidence in an exercise professional’s ability to provide suitable care for the referred patient, and for the exercise referral process to be trustworthy and effective.

Most health care providers report that greater access to *affordable* community PAE programs is needed in Nova Scotia. For instance, designing more public programs for PAE introduction and providing affordable access to *qualified* exercise professionals (as outlined above) through these subsidized programs could substantially increase PAE participation [27]. Increasing the number of community facilities that employ *qualified* exercise professionals would provide more opportunities to offer instructional classes, community programs, and PAE promotional tools (e.g., information graphics, school outreach programs, professional development sessions) to individuals at risk for, or with, medical conditions. With greater accessibility to *qualified* exercise professionals who run community-based programming, other health care professionals may be more inclined to refer their clients consistently, especially those clients with medical conditions. To make ‘active living’ more commonplace and to complement community-based physical activity and exercise programming, policies for the design of neighborhoods, workspaces, and schools must be a consideration going forward to support physical activity across a continuum [28]. Creating more urban green spaces, active transportation routes (e.g., bike lanes, walking trails), easily accessible community recreation centers, and PAE spaces in schools (e.g., playgrounds, gym facilities) should be a priority for policymakers [28].

Unique to their profession, physicians most frequently cited “changes to billing structure” because of the way the current billing structure incentivizes written medicinal prescriptions over PAE counselling. Through brief PAE counselling and prescription, the number needed to treat to have 1 patient successfully meet the weekly PAE aerobic guidelines of 150 min or more of moderate-to-vigorous physical activity, is 12 [29], making it comparatively effective to medications [4]. For context, the number needed to treat for smoking cessation is approximately 52 [30]. Previous reports point to lack of time, inexperience in providing quality PAE advice, patients’ lack of enthusiasm, and low confidence levels in PAE counselling [8,10]. Our findings suggest changes to billing structure are also important to physicians in Nova Scotia. Nevertheless, if organizational changes are implemented to provide a transparent, well-defined, and appropriately compensated billing structure for PAE, physicians may be more inclined to spend more time encouraging patients to engage in PAE as part of their clinical care.

Nurses cited “reducing patient financial barriers” as the primary solution to promoting and sustaining PAE in health care. It is unclear why nurses reported this more often than other health care providers, but it could be due to their frequent, prolonged direct patient interaction. Nonetheless, outlining more affordable PAE programs to patients is a point that nurses are looking for as it is within their scope to provide patients with preventative treatment options [31]. In contrast, only 11% dietitians cited reducing financial barriers as an important factor. Rather, 40% of dietitians described the importance of more educational opportunities for health care providers and 28% advocated for greater accessibility to *qualified* exercise professionals. This is likely due to dietitians training in behavior change strategies for proper nutrition, positioning them well to apply such principles to changing physical activity behaviors. Going forward, health care settings need to collaboratively communicate the importance of engaging in PAE and implement effective strategies to boost PAE adherence among patients.

The current study focused on PAE, but this represents one pillar of health. Health care providers are tasked with improving the overall health of their patients/clients, which may include PAE, diet, mental health, etc. Many of these providers address the overall health of their patients/clients using a more holistic perspective that includes these additional non-activity-related health factors. A better understanding of health care provider beliefs regarding the integration of other aspects of health are warranted. Respondents to a physical activity survey may have been more likely to be in favor of PAE promotion than non-respondents, although rates of PAE promotion were relatively low (Table 1). Also, 43% of survey respondents did not respond to the open-ended question that was coded for thematic content, which likely further reduced the representativeness. This was a survey of health care providers working in Nova Scotia. Therefore, our results are specific to this province and may not be representative of all Canadian health care providers. However, Nova Scotia has been shown to have higher rates of chronic disease and a strained health care system compared to the rest of Canada, emphasizing the need for increased PAE prescription [5]. It is worth noting that the survey data was collected prior to the coronavirus pandemic, which has added an additional barrier to including PAE content and has exacerbated rates of patient/client physical inactivity [32].

The study may be limited by its inclusion of an open-ended question survey versus a detailed, qualitative interview, but our large sample size strengthens our findings. Also, the open-ended question design allowed for respondents to provide their unprompted perspectives. Including large groups of different health care providers (i.e., physicians, exercise professionals, nurses, and dietitians) helped to identify the consensus of opinions across health care settings.

## 5. Conclusions

Through open-ended responses, approximately 25% of surveyed health care providers identified the need for more educational opportunities to increase their PAE knowledge, improve their self-confidence with providing PAE content, and improve their ability to overcome impactful barriers that prevent regular PAE promotion in Nova Scotia. Health care providers in Nova Scotia also believe that a greater availability of community-based PAE programs that are affordable and incorporate *qualified* exercise professionals, would support the promotion of PAE as a primary aspect of health care.

## Figures and Tables

**Figure 1 ijerph-19-09466-f001:**
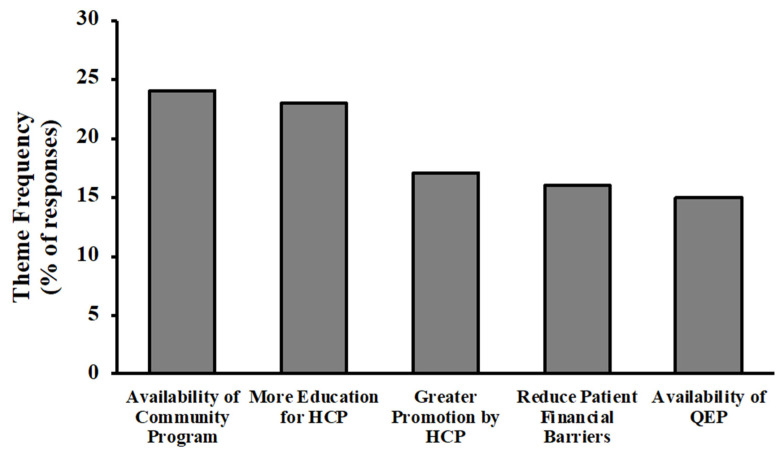
The most common themes regarding what could be done to promote and sustain physical activity and exercise in health care among the pooled sample of health care providers across Nova Scotia (*n* = 341; 1.7 responses per provider). HCP, health care professional; QEP, qualified exercise professional.

**Figure 2 ijerph-19-09466-f002:**
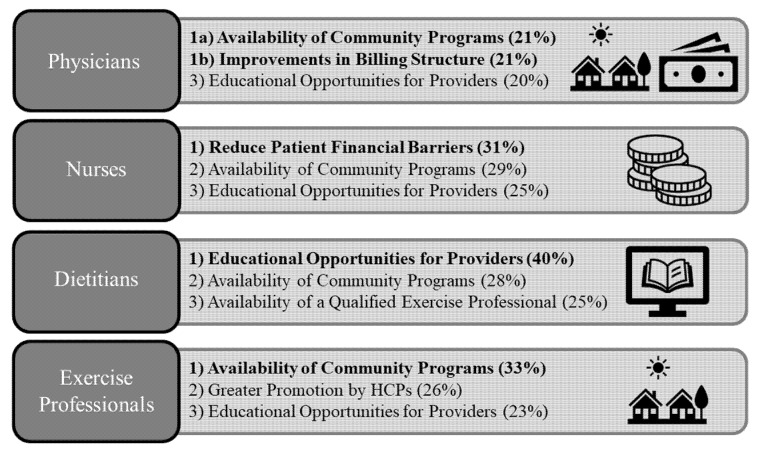
Summary of top 3 themes that physicians, nurses, dietitians, and exercise professionals working in Nova Scotia believe could be done to promote and sustain physical activity and exercise in health care. Each provider % (frequency of theme ÷ number of providers).

**Table 1 ijerph-19-09466-t001:** Participant characteristics and physical activity counselling or exercise prescription practices across physicians, nurses, dietitians, and exercise professionals.

	Physicians (*n* = 114)	Nurses (*n* = 48)	Dietitians (*n* = 65)	Exercise Professionals (*n* = 114)
Age (years)	52 ± 12	50 ± 10	41 ± 11	43 ± 11
Years of Practice	23 ± 13	27 ± 9	15 ± 10	18 ± 11
Gender (% women)	54.9	97.9	96.9	75.2
Race (% Caucasian)	86.7	97.9	100.0	94.6
Include PAE in sessions (% of patients/clients)	43 ± 30	55 ± 36	57 ± 30	85 ± 22
Assess PAE Participations (% of patients/clients)	47 ± 34	66 ± 37	60 ± 37	82 ± 26
Assess PAE Readiness (% of patients/clients)	35 ± 32	58 ± 38	57 ± 35	69 ± 35
Recommend PAE (% of patients/clients)	63 ± 30	73 ± 31	70 ± 28	93 ± 13
Prescribe Exercise (% of patients/clients)	13 ± 21	19 ± 30	20 ± 29	83 ± 24
Provide PAE Referral (% of patients/clients)	10 ± 18	17 ± 27	16 ± 22	25 ± 26

Data presented as frequency (%) or mean ± standard deviation. PAE, physical activity and exercise.

**Table 2 ijerph-19-09466-t002:** A detailed proportion of each theme across different health care providers in response to what they believe could be done to promote and sustain physical activity and exercise in health care.

Theme	Physicians (*n* = 114)	Nurses (*n* = 48)	Dietitians (*n* = 65)	Exercise Professionals (*n* = 114)
Community Program Availability (%)	21	29	28	33
Educational Opportunities for HCPs (%)	20	25	40	23
More Frequent Promotion by HCP (%)	12	22	17	26
Qualified Exercise Professional Availability (%)	11	12	25	20
Reduce Patient Financial Barriers (%)	19	31	11	16
Tools/Resources for HCPs to use (%)	12	12	17	9
Greater Public Promotion (%)	18	12	3	12
Improvements in Billing Structure (%)	21	4	2	8
Intervene in Childhood (%)	11	10	3	14
Tools/Resources for Patients (%)	10	10	5	8
Other (%)	9	10	9	8

Data presented as the frequency of each theme (frequency of theme ÷ number of each health care professional). HCP, health care professional.

## Data Availability

Not applicable.

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
