# Peer review of "Health Care Providers’ Perspectives on Promoting Physical Activity and Exercise in Health Care"

_ijerph, 2022, doi:10.3390/ijerph19159466_

Round 1
Reviewer 1 Report
Overall Comments:
This study reports on thematic content of the responses of health care providers and professionals engaged in nutrition/physical activity behavioral health professions. The one survey item reported in the results was an open-ended question about what can be done to sustain physical activity/exercise in health care. Themes were identified in all responses by two reviewers, and the proportions of respondents indicating each theme was tabulated. Frequencies of themes were then reported by health care provider type. This study is interesting, and with some additional information about the survey, and further engagement with the limitations of the study will provide valuable information about how to encourage health care providers and nutrition/physical activity behavioral health professionals to more consistently promote physical activity to patients and clients.
Specific remarks:
1. The study objective (“ascertain perspectives… to increase PAE promotion and prescription” lines 73-75) does not align well with the coded question (“what could be done to promote and sustain physical activity and exercise in health care?” lines 116-117). The objective is how to promote/sustain PAE promotion, but the question the health care providers responded to is about the activity itself (with no reference to promotion/prescription). It is possible that the differences that emerge between the different categories of professionals have different understandings of the question giving rise to different thematic patterns. This needs to be engaged with directly, either by refining the study objective or as a limitation.
2. Table 1: is patients the correct word to use for dieticians and exercise professionals? It seems like clients might be a more appropriate word for recipients of services rendered in a non-clinical setting.
3. Table 1, lines 133-134: the word prescription likely means very different things to different service providers. For physicians, prescription is a regulated service for providing authorization for patients to receive regulated pharmaceutical products. For other providers who do not generally prescribe regulated pharmaceutical products, prescription may have a very different meaning (more interchangeable with counseling, recommending). This should likely be addressed in the discussion.
4. Lines 185-186: reduction in all cause mortality needs a specified time period- 100% of people still die. Over what period of time was mortality reduced (e.g. X number of years after prescription).
5. Line 217: “suitable patient care” doesn’t seem appropriate as they should likely be referred to as clients, not patients, in a community setting. If the “patient care” is in reference to the HCP, it should be changed to “suitable care for the referred-patient”.
6. Lines 222-223: claim about substantial increase in PAE participation through subsidized community programs needs to have a citation or be removed.
7. Lines 238-239: number needed to treat needs to be described as “the number needed to treat to have 1 patient successfully meet the weekly PAE aerobic guidelines…”.
8. Line 242: the NNT for smoking cessation is not 1 in 52, it is 52.
9. Lines 262-264: the sample included in this analysis also excluded 43% of survey respondents who did not respond to the open-ended question that was coded for thematic content. This likely further reduces the representativeness.
Reviewer 2 Report
My suggestion is in the attachement.
